

# Validation of an aspiration risk prediction model for Parkinson's disease based on nomogram: a single-center study

Yan Yan Xu[1,*], Yun Wei[2,*] and Ling Sha[1]

[1] Nursing Division of the Department of Neurology, Nanjing Drum Tower Hospital Affiliated to Nanjing University Medical School, Nanjing, Jiangsu, China
[2] Nursing Division of the Department of Cardiology, Nanjing Drum Tower Hospital Affiliated to Nanjing University Medical School, Nanjing, Jiangsu, China
[*] These authors contributed equally to this work.

Corresponding authors
Yan Yan Xu, 18915952477@163.com
Ling Sha, sani666@sina.com, lingsha025@163.com

## ABSTRACT

**Objective**. This study aims to develop and validate a predictive model for aspiration risk in patients with Parkinson's disease (PD).

**Methods**. A total of 160 inpatients with PD were enrolled (December 2022 to December 2023) from the Neurology Department of the Affiliated Drum Tower Hospital. Of 33 candidate variables, univariate analysis and Least Absolute Shrinkage and Selection Operator (LASSO) logistic regression were used to identify key predictors and construct a clinical nomogram. Model discrimination and calibration were assessed using receiver operating characteristic (ROC) curves and calibration plots.

**Results**. Univariate analysis and LASSO regression reduced the 33 variables to four core predictors: history of choking cough (odds ratio (OR) = 11.427; 95% confidence interval (CI) [2.187–59.709]), abnormal water-swallowing test results (OR = 4.262, 95% CI [1.496–12.140]), reduced facial expression (OR = 2.929, 95% CI [1.055–8.134]), and Barthel Index (OR = 0.972, 95% CI [0.950–0.995]). The area under the curve (AUC) values of the model were 0.882 (optimism-adjusted) and 0.950 for the training and testing sets, respectively. Calibration and decision curve analyses further validated the high performance and clinical utility of this model.

**Conclusion**. This nomogram effectively stratified aspiration risk in patients with PD, facilitating earlier detection and intervention. Future studies including more clinical variables and larger multicenter cohorts are required to enhance the predictive accuracy and generalizability of the model.

## INTRODUCTION

Parkinson's disease (PD), a neurodegenerative disorder, has attracted increasing attention. It is primarily characterized by the progressive loss of dopaminergic neurons, resulting in both motor and non-motor symptoms (*Mhyre et al., 2012*). Patients with PD often experience a range of non-motor symptoms such as depression, anxiety, and sleep

disturbances, which significantly impair their quality of life (*Lee & Koh, 2015*). Dysphagia is a common clinical symptom in PD, with reports indicating that over 80% of patients develop this condition during the disease course, and nearly all experience it in advanced stages (*Frank et al., 2021*; *Simons et al., 2014*; *Suttrup & Warnecke, 2016a*). Dysphagia markedly increases the risk of aspiration, leading to aspiration pneumonia, which accounts for approximately 70% of deaths in patients with PD (*Frank et al., 2021*). Therefore, early identification of aspiration risk is crucial for preventing complications and improving patient outcomes.

Clinically, fiberoptic endoscopic evaluation of swallowing (FEES) is considered the gold standard for assessing swallowing function. Previous studies have compared subjective perceptions of swallowing in patients with PD with objective assessments obtained from FEES or video fluoroscopic swallowing studies (VFSS). These studies demonstrated that over 50% of patients with PD who subjectively reported no swallowing disorders exhibited swallowing impairments on objective testing, with up to 15% experiencing silent aspiration (*Suttrup & Warnecke, 2016b*). However, VFSS has several limitations: it requires specialized equipment, requires the transfer of patients to the radiology department for examination, demands accurate assessment by trained professionals, and exposes patients to radiation. Consequently, the widespread application of VFSS is limited.

Currently, simple methods commonly used in clinical practice to assess the risk of aspiration include the Swallowing Disturbance Questionnaire and the Munich Dysphagia Test-Parkinson's Disease (MDT-PD), which serve as initial screening tools for swallowing disorders in patients with PD. These instruments have reported sensitivities of 80.5% and 81.3%, and specificities of 82% and 71%, respectively (*Manor et al., 2007*; *Simons et al., 2014*). However, they are primarily administered by clinicians and are not integrated into nursing assessment protocols. Moreover, when dysphagia symptoms are not prominent, medical and nursing staff may overlook them. As a result, early assessments are not effectively implemented, and aspiration may only be diagnosed after severe complications have arisen, delaying optimal intervention timing.

Although previous studies have explored aspiration risk in patients with PD, current assessment methods still predominantly rely on clinical experience and basic screening tools (*Troche et al., 2016*). However, these methods lack standardization and precision, which significantly limits their effectiveness in clinical practice. The existing literature suggests that specific phenotypic features—such as a history of choking and coughing, results of the water-swallowing test, and abnormal facial expressions—are closely associated with an increased risk of aspiration (*El Halabi, Arwani & Parkman, 2023*; *Fasseeh et al., 2021*; *Mari et al., 1997*; *Stewart et al., 2024*). To date, however, no studies have systematically integrated these factors to create a comprehensive predictive model. To address this need, our study aimed to develop and validate a predictive model for aspiration risk in patients with PD, facilitating early identification and intervention to ultimately enhance their quality of life.

## MATERIALS AND METHODS

### Study population

This study included patients diagnosed with PD who were admitted to the Neurology Department of the Affiliated Drum Tower Hospital, Medical School of Nanjing University, between December 2022 and December 2023. The inclusion and exclusion criteria for participant selection were as follows:

### Inclusion criteria

(i) Confirmed diagnosis of PD based on established diagnostic criteria (*Postuma et al., 2018*).

(ii) Complete clinical data, including demographic information and relevant clinical assessments.

(iii) Clear diagnostic evidence for patients with aspiration.

(iv) Patients must be conscious and capable of oral intake.

### Exclusion criteria

(i) Incomplete clinical records.

(ii) Patients with pre-existing aspiration at the time of hospital admission

(iii) Patients with other conditions that could independently cause aspiration (*e.g.*, stroke) or with severe illnesses (*e.g.*, urosepsis or multi-organ failure).

(iv) Length of hospitalization less than three days.

### Diagnosis criteria

The criteria for diagnosing aspiration were defined in accordance with the protocol established by *Jaillette et al. (2015)* for the randomized controlled trial. Specifically, a pepsin A concentration >25 ng/mL in respiratory secretions was used as the diagnostic threshold for confirming aspiration, thereby excluding general hospital-acquired pneumonia. In this study, the occurrence of aspiration within 3 days of hospital admission, with exclusion of pre-existing aspiration, was considered the clinical endpoint for developing a predictive model. All baseline clinical measurements were performed within 24 h of admission.

Informed consent for this study was obtained *via* telephone. Participants were provided with detailed information regarding the study's purpose, procedures, and potential risks, and their consent was verbally recorded. The study was approved by the Institutional Review Board (IRB) of Nanjing University Affiliated Drum Tower Hospital (IRB number: 2024-779-01).

### Statistical analysis and model construction

Continuous variables with a normal distribution are expressed as mean $\pm$ standard deviation, whereas those with a non-normal distribution are reported as median and interquartile range. Comparisons between groups were made using Student's $t$-test for normally distributed variables with homogeneous variances and the Mann–Whitney U test for non-normally distributed variables or those with heterogeneous variances. Categorical variables were analyzed using the chi-square test to identify significant differences between patients with and without aspiration pneumonia.

All statistical analyses were performed using R software (version 4.4.2). The dataset was randomly divided into a training set (75%, $n = 120$) and a testing set (25%, $n = 40$). Within the training set, univariate analysis and LASSO logistic regression with 10-fold cross-validation were employed sequentially to identify significant predictive factors from the 33 candidate variables and build the final model. Variables with $p < 0.01$ in univariate analysis were first selected and entered into the LASSO logistic regression. The non-zero coefficients of the trained LASSO model were directly used for constructing the nomogram. The "car" package in R was used to calculate the variance inflation factor (VIF) to assess multicollinearity among the selected variables. The optimal cutoff value for the model was determined using the Youden index. Model performance was evaluated in the training and testing sets, focusing on metrics including the area under the receiver operating characteristic (ROC) area under the curve (AUC), sensitivity, specificity, positive predictive value, negative predictive value, positive likelihood ratio, negative likelihood ratio, true positives, false positives, false negatives, true negatives, accuracy, Brier score, calibration curves, and decision curve analysis. Bootstrapping with 1,000 iterations was applied to the training set to adjust for optimism in performance and to the testing set to obtain 95% confidence intervals (95% CI). All statistical tests were two-tailed, with a significance threshold of $p < 0.05$.

## RESULTS

### Characteristics of the study cohort

Significant differences were observed in several demographic and clinical parameters between the aspiration and non-aspiration groups ($N = 80$ in each group). The aspiration group had a higher median age of 74 years (IQR, 67–77) compared with 66 years (IQR, 58.5–70.5) in the non-aspiration group ($p < 0.001$). The proportion of male patients was higher in the aspiration group (76.25%) than in the non-aspiration group (61.25%; $p = 0.045$). Patients with aspiration had a median hospital length of stay of 12 days (IQR, 8–18) compared with 7 days (IQR: 5–10) for those without aspiration ($p < 0.001$). A markedly higher prevalence of a history of choking cough was observed in the aspiration group (55.00%) than in the non-aspiration group (2.50%; $p < 0.001$). Abnormal facial expressions were noted more frequently in the aspiration group (71.25%) than in the non-aspiration group (26.25%; $p < 0.001$). Similarly, abnormal findings on the water-swallowing test were more common in the aspiration group (81.25% *vs.* 21.25%; $p < 0.001$).

The aspiration group exhibited significantly higher rates of bradykinesia (97.50% *vs.* 82.50%, $p = 0.004$) and tremor (83.75% *vs.* 63.75%, $p = 0.007$) than the non-aspiration group. A substantially greater proportion of patients in the aspiration group had severe disease (68.75%) compared to the non-aspiration group (17.50%; $p < 0.001$). Additionally, abnormal cough reflexes were observed more frequently in the aspiration group (35.00% *vs.* 2.50%; $p < 0.001$). Nutritional risk, assessed using the 2002 Nutritional Risk Screening Scale, was significantly higher in the aspiration group (51.25%) compared to the non-aspiration group (21.25%; $p < 0.001$) (Table 1).

**Table 1  Baseline characteristics of the non-aspiration and aspiration groups.**

|  | Non-Aspiration (N = 80) | Aspiration (N = 80) | p value |
|---|---|---|---|
| Gender |  |  | 0.061 |
| Female | 31 (38.75%) | 19 (23.75%) |  |
| Male | 49 (61.25%) | 61 (76.25%) |  |
| Age (years old) | 66.00 [58.50; 70.50] | 74.00 [67.00; 77.00] | <0.001 |
| HospitDur (days) | 7.00 [5.50; 9.00] | 10.00 [8.00; 12.00] | <0.001 |
| Barthel | 85.00 [65.00; 95.00] | 59.00 [41.00; 77.50] | <0.001 |
| DiseDur (years) | 2.00 [1.00; 4.00] | 4.15 [1.90; 9.20] | <0.001 |
| NutriSt |  |  | 0.001 |
| No Emaciation | 59 (73.75%) | 37 (46.25%) |  |
| Emaciation | 21 (26.25%) | 43 (53.75%) |  |
| AlcoHist |  |  | 0.034 |
| No | 72 (90.00%) | 79 (98.75%) |  |
| Yes | 8 (10.00%) | 1 (1.25%) |  |
| SmokeHist |  |  | <0.001 |
| No | 64 (80.00%) | 79 (98.75%) |  |
| Yes | 16 (20.00%) | 1 (1.25%) |  |
| MariSt |  |  | 0.083 |
| Unmarried | 3 (3.75%) | 10 (12.50%) |  |
| Married | 77 (96.25%) | 70 (87.50%) |  |
| BradyK |  |  | 0.004 |
| No | 14 (17.5%) | 2 (2.50%) |  |
| Yes | 66 (82.50%) | 78 (97.50%) |  |
| Tremor |  |  | 0.007 |
| No | 29 (36.25%) | 13 (16.25%) |  |
| Yes | 51 (63.75%) | 67 (83.75%) |  |
| DiseSev |  |  | <0.001 |
| Hoehn-Yahr ≤ 3 | 66 (82.50%) | 25 (31.25%) |  |
| Hoehn-Yahr ≥ 4 | 14 (17.50%) | 55 (68.75%) |  |
| Salivation |  |  | 0.083 |
| No | 77 (96.25%) | 70 (87.50%) |  |
| Yes | 3 (3.75%) | 10 (12.50%) |  |
| CoughRef |  |  | <0.001 |
| No | 78 (97.50%) | 52 (65.00%) |  |
| Yes | 2 (2.50%) | 28 (35.00%) |  |
| SpeechInd |  |  | 0.082 |
| No | 68 (85.00%) | 58 (72.50%) |  |
| Yes | 12 (15.00%) | 22 (27.50%) |  |
| CogImp |  |  | 0.422 |
| No | 50 (62.50%) | 44 (55.00%) |  |
| Yes | 30 (37.50%) | 36 (45.00%) |  |

| | Non-Aspiration (*N* = 80) | Aspiration (*N* = 80) | *p* value |
|---|---|---|---|
| DeprSt | | | 0.444 |
| No | 60 (75.00%) | 65 (81.25%) | |
| Yes | 20 (25.00%) | 15 (18.75%) | |
| ChokCghHist | | | <0.001 |
| No | 78 (97.50%) | 36 (45.00%) | |
| Yes | 2 (2.50%) | 44 (55.00%) | |
| AppetLoss | | | <0.001 |
| No | 74 (92.50%) | 55 (68.75%) | |
| Yes | 6 (7.50%) | 25 (31.25%) | |
| WeightLoss | | | 0.532 |
| No | 76 (95.00%) | 73 (91.25%) | |
| Yes | 4 (5.00%) | 7 (8.75%) | |
| MemDecl | | | 1 |
| No | 71 (88.75%) | 70 (87.50%) | |
| Yes | 9 (11.25%) | 10 (12.50%) | |
| PulmDis | | | 0.818 |
| No | 70 (87.50%) | 68 (85.00%) | |
| Yes | 10 (12.50%) | 12 (15.00%) | |
| CardCerebDis | | | 0.196 |
| No | 53 (66.25%) | 44 (55.00%) | |
| Yes | 27 (33.75%) | 36 (45.00%) | |
| FaceExpr | | | <0.001 |
| No | 59 (73.75%) | 23 (28.75%) | |
| Yes | 21 (26.25%) | 57 (71.25%) | |
| ConstHist | | | 0.017 |
| No | 43 (53.75%) | 27 (33.75%) | |
| Yes | 37 (46.25%) | 53 (66.25%) | |
| Pain | | | 1 |
| No | 71 (88.75%) | 72 (90.00%) | |
| Yes | 9 (11.25%) | 8 (10.00%) | |
| NRSScale | | | <0.001 |
| NRS2002 <2 | 63 (78.75%) | 39 (48.75%) | |
| NRS2002 ≥ 2 | 17 (21.25%) | 41 (51.25%) | |
| WaterSwaT | | | <0.001 |
| Normal | 63 (78.75%) | 15 (18.75%) | |
| Abnormal | 17 (21.25%) | 65 (81.25%) | |
| WBC ($\times 10^9$/L) | 5.35 [4.65; 6.65] | 5.70 [4.65; 7.45] | 0.118 |
| HB (g/L) | 135.00 [122.00; 146.50] | 132.00 [122.05; 136.35] | 0.076 |

|  | Non-Aspiration ($N = 80$) | Aspiration ($N = 80$) | *p* value |
|---|---|---|---|
| CRP (mg/L) | 2.60 [1.85; 4.70] | 4.95 [2.95; 14.50] | <0.001 |
| Lym ($\times 10^9$/L) | 1.60 [1.30; 1.95] | 1.50 [1.00; 2.00] | 0.182 |
| Neu ($\times 10^9$/L) | 3.20 [2.60; 4.30] | 3.60 [3.00; 4.40] | 0.024 |

**Notes.**

Abbreviations: HospitDur, hospitalization duration; Barthel, Barthel Index; DiseDur, disease duration; NutriSt, nutritional status; AlcoHist, alcohol history; SmokeHist, smoking history; MariSt, marital status; BradyK, bradykinesia; DiseSev, disease severity; CoughRef, cough reflex; SpeechInd, speech indistinct; CogImp, cognitive impairment; DeprSt, depression status; ChokCghHist, choking cough history; AppetLoss, loss of appetite; MemDecl, memory decline; PulmDis, pulmonary disease; CardCerebDis, cardiovascular cerebral disease; FaceExpr, facial expression; ConstHist, constipation history; NRSScale, Nutritional Risk Screening 2002 scale; WaterSwaT, water-swallowing test; WBC, white blood cell; HB, hemoglobin; CRP, C-reactive protein; Lym, lymphocyte; Neu, neutrophil.

DiseDur, number of years of having PD before admission; NutriSt, emaciation is defined as BMI<18.5 kg/m$^2$, which is a standard international classification criterion for underweight status by WHO; AlcoHist, yes if consumed alcohol regularly or irregularly for at least one year or longer in the past; SmokeHist, yes if having a history of regularly smoking for at least six consecutive months or longer; BradyK, yes if the Movement Disorder Society-Unified Parkinson's Disease Rating Scale (MDS-UPDRS) >= 1 on limb-based motor tasks; Tremor, yes if MDS-UPDRS Tremor Score >= 1; Salivation, yes if incapable to voluntarily control the flow of saliva out of the mouth, requiring frequent wiping; CoughRef, yes if having rapid and intense protective coughing reflex when a foreign object or irritant enters the trachea. SpeechInd, yes if exhibiting slurred and effortful speech, abnormal pitch or rhythm, confirmed through clinical observation or speech assessment tools; CogImp, yes if total scores on the MMSE (Mini-Mental State Examination) or MoCA (Montreal Cognitive Assessment) scales fall below the normal thresholds (MoCA <26 points or MMSE <27 points); DeprSt, determined by a clinical assessment combined with professional scales (such as a HAMD depression scale score ≥7 or a PHQ-9 score ≥5), or by clear documented medical history; ChokCghHist, yes if having a history of clearly documented or self-reported experiences of coughing or choking caused by food, liquid, or medication entering the airway during swallowing; AppetLoss, yes if significant reduction in food intake before the onset of illness, or recent (within one month) food intake significantly below 50% of regular intake, lasting for one week or longer; WeightLoss, yes if having unintentional weight loss exceeding 5% of the original body weight within the past 3 to 6 months; MemDecl, yes if self or family reporting a noticeable recent decline in memory, and having abnormal and decreased memory scores on the MoCA or MMSE memory items; PulmDis, yes if having clinically confirmed pneumonia, chronic bronchitis, COPD (chronic obstructive pulmonary disease), lung infections, or other pulmonary diseases; CardCerebDis, yes if having clinically confirmed coronary heart disease, hypertension, cerebral infarction, cerebral hemorrhage, transient ischemic attack, or history of cerebrovascular disease; FaceExpr, yes if exhibiting no significant facial expression changes, presenting with a blank expression, lack of facial animation, and reduced blinking. ConstHist, yes if having reduced frequency of bowel movements (≤3 times per week), dry and hard stools, difficulty in defecation; Pain, yes if self-reporting pain in any body part, with recorded pain hisotry, or confirmed pain score of ≥1 using pain assessment scales such as the Visual Analogue Scale or Numerical Rating Scale (NRS). WaterSwaT, abnormal if coughing or difficulty swallowing occurs during a standardized water swallowing test.

## Results of predictor selection and model construction

In the training set, univariate analysis initially identified 11 potential predictors from the 33 candidate variables. Subsequent LASSO logistic regression yielded four final predictors: ChokCghHist (choking cough history, OR = 11.427, 95% CI [2.187–59.709]; $P = 0.004$), WaterSwaT (water-swallowing test, OR = 4.262, 95% CI [1.496–12.140]; $P = 0.007$), FaceExpr (facial expression, OR = 2.929, 95% CI [1.055–8.134]; $P = 0.039$), and Barthel (Barthel Index, OR = 0.972, 95% CI [0.950–0.995]; $P = 0.017$). These emerged as independent factors differentiating aspiration status. Because all VIF values were below 5, severe multicollinearity was ruled out (Table 2).

A nomogram was constructed based on these four variables to distinguish aspiration status (Fig. 1) by directly using the LASSO logistic regression coefficients. To estimate aspiration risk, the levels of ChokCghHist, WaterSwaT, FaceExpr, and Barthel were evaluated, and individual scores were assigned according to the nomogram's scale. The sum of these scores yielded a total score from which the probability of aspiration could be inferred.

**Table 2  LASSO logistic regression analysis in the training set.**

| Variable | VIF | β | OR (95% CI) | p value |
| --- | --- | --- | --- | --- |
| ChokCghHist | 2.152 | 2.436 | 11.427 (2.187–59.709) | 0.004 |
| FaceExpr | 2.192 | 1.075 | 2.929 (1.055–8.134) | 0.039 |
| WaterSwaT | 2.609 | 1.450 | 4.262 (1.496–12.140) | 0.007 |
| Barthel | 1.670 | −0.028 | 0.972 (0.950–0.995) | 0.017 |

**Notes.**

Abbreviations: ChokCghHist, choking cough history; FaceExpr, facial expression; WaterSwaT, water-swallowing test; VIF, variance inflation factor.

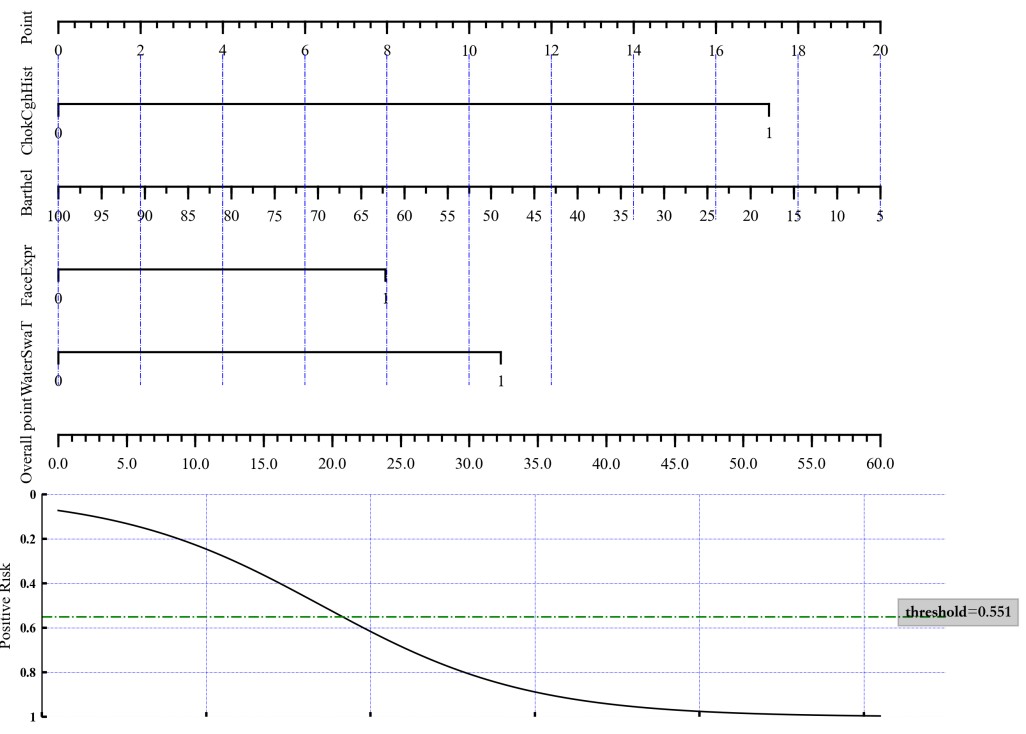

**Figure 1  Nomogram of the LASSO logistic regression model for discrimination between aspiration and non-aspiration.**

## Model performance

We estimated the probability of three-day aspiration occurrence for each patient as the output of the LASSO logistic regression model and assessed performance using ROC curves. As shown in Fig. 2 and Table 3, the AUC for the training set was 0.891 and 0.882 (95% CI [0.832–0.945]) after optimism adjustment by bootstrapping. The testing AUC was 0.950 (95% CI [0.872–1.000]), highlighting the model's robust discriminative ability based on the parameters ChokCghHist, WaterSwaT, FaceExpr, and Barthel. The optimal cutoff score, determined using the Youden index, was 0.551. For the training and testing sets, sensitivity (95% CI) was 0.75 (0.58–0.89, adjusted) and 0.80 (0.61–0.95), respectively, while specificity (95% CI) was 0.89 (0.83–0.97, adjusted) and 1.00 (1.00–1.00), respectively. Correspondingly, accuracy (95% CI) reached 0.82 (0.73–0.91, adjusted) for

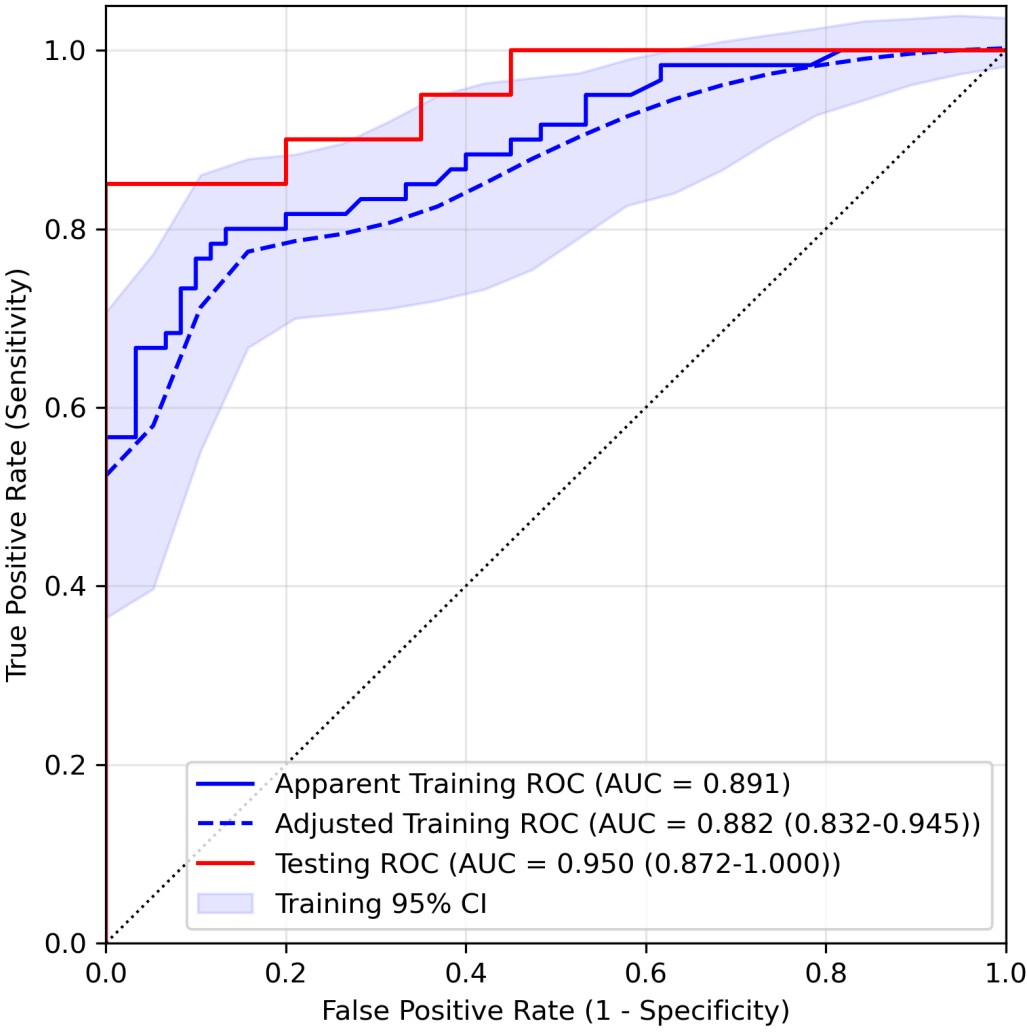

**Figure 2** **ROC curve of the LASSO logistic regression model in the training set and testing set.** Both the apparent and optimism-adjusted training ROCs (with 95% confidence interval) were drawn in the figure.

the training cohort and 0.90 (0.80–0.97) for the testing set, with Brier scores (95% CI) of 0.138 (0.101–0.176, adjusted) and 0.091 (0.052–0.138), respectively.

The calibration curve (Fig. 3) showed that the predicted probabilities generated by the LASSO logistic nomogram closely approximated the actual incidence of aspiration, indicating a high level of predictive accuracy. Furthermore, decision curve analysis (Fig. 4) confirmed the clinical utility of the model across both the training and external testing sets. In the training set, when the probability threshold for intervention ranged from 20% to 90%, the model's net benefit substantially surpassed the conventional extremes of treating all patients or withholding treatment entirely (Fig. 4A). These findings were corroborated

**Table 3  Accuracy of the prediction score of the LASSO logistic regression model for discrimination between aspiration and non-aspiration.**

| Variable | Training set (adjusted, 95% CI) | Testing set (95% CI) |
|---|---|---|
| AUC | 0.882 (0.832–0.995) | 0.950 (0.872–1.000) |
| Cutoff probability | 0.551 | 0.551 |
| Sensitivity | 0.749 (0.581–0.891) | 0.800 (0.611–0.952) |
| Specificity | 0.894 (0.833–0.972) | 1.000 (1.000–1.000) |
| PPV | 0.873 (0.803–0.957) | 1.000 (1.000–1.000) |
| NPV | 0.785 (0.687–0.887) | 0.833 (0.680–0.960) |
| PLR | 2.073 (0–13.865) | Not estimable |
| NLR | 0.278 (0.118–0.446) | 0.200 (0.048–0.389) |
| TP | 46 | 16 |
| FP | 6 | 0 |
| FN | 54 | 20 |
| TN | 14 | 4 |
| Accuracy | 0.820 (0.733–0.908) | 0.900 (0.800–0.975) |

**Notes.**

Abbreviations: PPV, Positive Predictive Value; NPV, Negative Predictive Value; PLR, Positive Likelihood Ratio; NLR, Negative Likelihood Ratio; TP, True Positive; FP, False Positive; FN, False Negative; TN, True Negative.

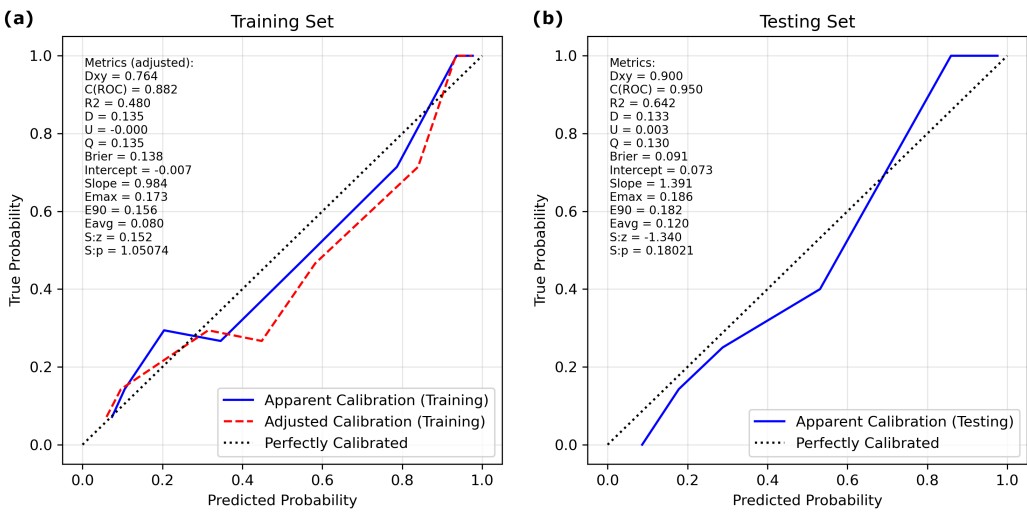

**Figure 3  Calibration curves of the LASSO logistic regression model for discrimination between aspiration and non-aspiration in the (A) training set and (B) validation set.**

by the external validation cohort, which exhibited a similarly elevated net benefit across a comparable range of intervention thresholds (Fig. 4B).

For example, at a 20% probability threshold—indicating that clinical intervention is considered once the risk of aspiration reaches this level—the model's net benefit in the training and testing sets was approximately 0.40 and 0.43, respectively. Both values markedly exceeded the net benefits observed under the treat-all (0.37) or treat-none (0) strategies, highlighting the practical applicability of the model in guiding patient management.

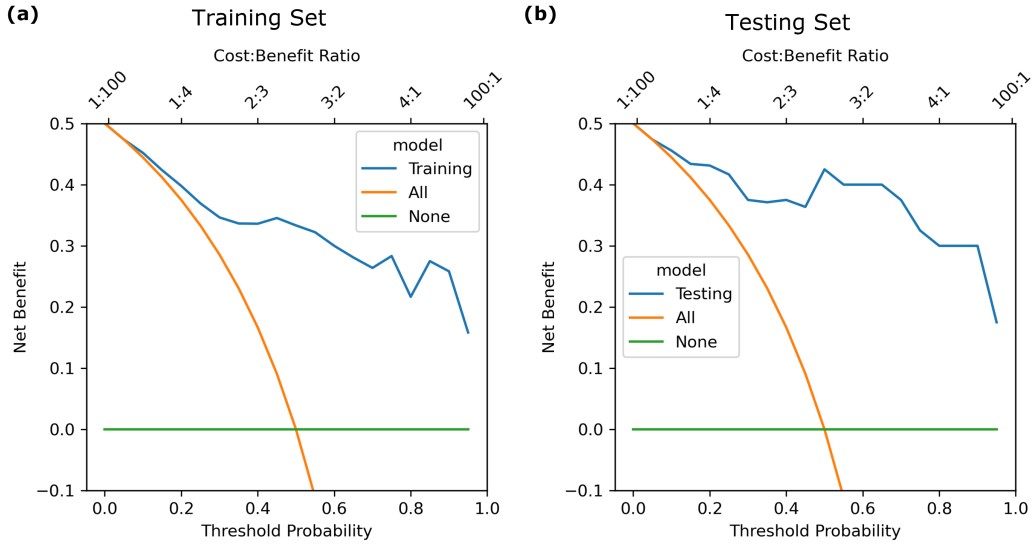

**Figure 4** Decision curve of the LASSO logistic regression model for discrimination between aspiration and non-aspiration in the (A) training set and (B) validation set.

Moreover, the model maintained stable and advantageous predictive performance even at higher probability thresholds (70%–90%), a range in which accurate risk stratification is particularly critical in stringent clinical scenarios. These outcomes suggest that this predictive tool not only enhances clinicians' ability to identify individuals at meaningful risk of aspiration but also provides an evidence-based framework to refine decision-making processes, ultimately supporting more targeted and resource-efficient patient care.

## DISCUSSION

PD is commonly characterized by bradykinesia, reduced coordination, and slower movements as its primary clinical features. However, the precise mechanisms underlying swallowing dysfunction in patients with PD remain unclear. Some investigators have suggested that damage to both dopaminergic and non-dopaminergic pathways, which affect the central swallowing center and peripheral neuromuscular elements, contributes to the onset and progression of PD-related swallowing difficulties (*Suttrup & Warnecke, 2016b*). Impaired motor function in PD can profoundly disrupt the normal swallowing process, leading to reduced laryngeal elevation, prolonged pharyngeal and overall swallowing duration, delayed initiation of the swallowing reflex, and, ultimately, aspiration (*Ertekin et al., 2002*).

Swallowing difficulties in individuals with PD may develop gradually and can appear at any stage of the disease, regardless of severity. Even in early PD, 95%–100% of patients exhibit swallowing impairments, often in a subclinical form; however, these may go unnoticed by both patients and healthcare providers. Consequently, the association between PD and swallowing dysfunction is frequently overlooked at early stages. By the time patients recognize swallowing problems, many have already progressed to moderate

or severe impairment, thereby delaying a formal diagnosis. Notably, there is an apparent discrepancy between patients' subjective perceptions and objective assessment findings. One study (*Palmer et al., 2021*) showed that relying solely on patient self-reports yielded a prevalence of approximately 35%, whereas objective evaluation tools yielded a prevalence closer to 85%. Furthermore, a substantial proportion of these patients experience "hidden" swallowing issues (*Kalf et al., 2012*), and some may even encounter silent aspiration in the early stages of the disease. Collectively, these factors complicate clinical efforts to identify aspiration risk in patients with PD.

Existing research has identified age as an independent risk factor for swallowing dysfunction. Sex, body mass index, disease duration, disease severity, sialorrhea, depression, and cognitive status have also been linked to clinical swallowing problems (*Cereda et al., 2014*; *Ding et al., 2018*; *Frank et al., 2021*; *Nienstedt et al., 2019*; *Simons et al., 2014*). Additional studies have suggested that dysarthria, reduced cough reflex sensitivity, and decreased peak cough airflow may be correlated with swallowing impairment and aspiration (*Curtis & Troche, 2020*; *Hegland, Okun & Troche, 2014*).

In the present study, the four significant risk factors for aspiration in patients with PD were a history of choking and coughing, a high score on the water-swallowing test, diminished facial expression, and a low Barthel Index. Episodes of choking and coughing indicate prior aspiration events, implying overt swallowing dysfunction or compromised muscle strength and coordination, both of which increase the likelihood of recurrent aspiration as PD progresses. The water-swallowing test is commonly used to evaluate swallowing function by observing whether the patient coughs or chokes during water intake. A score of 3 or higher suggests an underlying swallowing problem that increases the risk of aspiration. Our findings also indicate that reduced facial expression is a risk factor for aspiration. In 1986, Charcot introduced the concept of the "mask face," proposing facial characteristics as a diagnostic criterion for PD (*Goetz, 1986*). This "mask face" represents slowed or diminished facial muscular movements, reducing the velocity, elasticity, and coordination of the eyebrows, eyes, cheeks, and lips, thereby affecting overall facial expressiveness. This is a hallmark motor symptom of PD (*Bologna et al., 2013*; *Maremmani et al., 2019*).

One proposed mechanism for facial bradykinesia involves widespread pathological changes and neuronal loss within the substantia nigra dopaminergic system, along with insufficient secretion of the inhibitory neurotransmitter dopamine. These alterations result in predominant excitatory neurotransmission, ultimately leading to motor dysfunction and increased muscle tone (*Frank et al., 2021*; *Simons et al., 2014*). During early-stage PD, presynaptic dopaminergic function in the putamen, orbitofrontal cortex, and amygdala is altered; dopamine transmission in the mesocortical pathway is impaired; and extensive neural changes throughout the brain contribute to deficits in facial emotion recognition (*Argaud et al., 2018*). Damage to related telencephalic structures also influences swallowing control. Functional MRI studies have shown that both the bilateral putamen and globus pallidus are activated during swallowing in healthy individuals; however, these regions remain inactive in patients with PD (*Suzuki et al., 2003*). Additionally, impaired

neuromuscular control may lead to muscle rigidity, causing slow movement, stiffness, tremors, and facial motor symptoms (*Wang & Tickle-Degnen, 2018*).

In our study, the Barthel Index emerged as an independent predictor of aspiration risk, highlighting its significance in evaluating functional independence among patients with PD. The Barthel Index is a widely recognized tool that assesses a patient's ability to perform essential activities of daily living, such as feeding, dressing, and mobility, with lower scores reflecting greater dependency (*Taghizadeh et al., 2020*). This functional measure is particularly relevant to aspiration risk, as patients with reduced independence in tasks such as eating or maintaining posture may struggle with swallowing coordination or managing oral secretions—key contributors to aspiration in PD (*Gong et al., 2022*; *Murcia et al., 2010*).

The clinical importance of this finding lies in its potential to guide risk stratification and intervention strategies. Our model demonstrated that lower Barthel scores were strongly associated with an elevated likelihood of aspiration, consistent with previous studies linking functional decline to adverse outcomes such as aspiration pneumonia. By incorporating the Barthel Index, our nomogram achieved enhanced predictive accuracy, suggesting that functional status is not merely a covariate but a critical determinant of aspiration risk in this population.

The primary limitation of this study was its small sample size, which may have impacted the generalizability of the findings. Additionally, because the study was conducted at a single medical institution, regional bias may have limited the broader applicability of the developed model and its conclusions. Furthermore, only baseline clinical measurements were considered for model development; dynamic changes in clinical measurements, as well as subsequent interventions such as changes in diet and route of feeding, may contribute to more precise prediction of aspiration. Finally, the study did not consider other potential inhalation risk factors such as oral hygiene, saliva control, anticholinergic burden, and cough-reflex sensitivity, which could further improve the model's accuracy if included as predictors in future multicenter studies.

## CONCLUSION

In conclusion, this study identified four independent risk factors for aspiration using multivariate logistic regression and developed a nomogram with strong predictive performance. The model exhibited high accuracy and stability across both training and testing datasets, highlighting its substantial clinical value. Future research should aim to expand the sample size, conduct multicenter validation, and incorporate additional clinical indicators to enhance the model's generalizability and predictive capacity, ultimately providing a more effective tool for clinical risk assessment.

## ACKNOWLEDGEMENTS

We thank Editage for its English language editing services.

### Funding

This study was supported by Nanjing Drum Tower hospital affiliated to Nanjing university medical school nursing research project (Grant number 2023-A951). The funders had no role in study design, data collection and analysis, decision to publish, or preparation of the manuscript.

### Grant Disclosures

The following grant information was disclosed by the authors:
Nanjing Drum Tower hospital affiliated to Nanjing university medical school nursing research project: 2023-A951.

### Competing Interests

The authors declare there are no competing interests.

### Author Contributions

- Yan yan Xu conceived and designed the experiments, performed the experiments, analyzed the data, prepared figures and/or tables, and approved the final draft.
- Yun Wei conceived and designed the experiments, performed the experiments, prepared figures and/or tables, and approved the final draft.
- Ling Sha conceived and designed the experiments, authored or reviewed drafts of the article, and approved the final draft.

### Human Ethics

The following information was supplied relating to ethical approvals (*i.e.*, approving body and any reference numbers):

This study received ethical approval from the Ethics Committee of Nanjing Drum Tower Hospital Affiliated to Nanjing University Medical School (2024-779-01).

### Data Availability

The raw data and code are available in the Supplemental Files.

### Supplemental Information

Supplemental information for this article can be found online at http://dx.doi.org/10.7717/peerj.20443#supplemental-information.

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
