# Peer review of "Validation of an aspiration risk prediction model for Parkinson’s disease based on nomogram: a single-center study"

_PeerJ, doi:10.7717/peerj.20443_

## Round 0.1 · original submission · Major Revisions

· Academic Editor

Major Revisions

·

Basic reporting

No comment

Experimental design

The authors present a presumably prospective clinical study that acquired multiple clinical data points to develop a predictive model of aspiration pneumonia in hospitalized patients with Parkinson's Disease (PD). It is clinically important to identify patients at particularly high risk of aspiration, in order to change diet (i.e., thickened liquids and/or NPO status with alternative enteral feeding). Thus, the authors should be congratulated on developing this model. The research question is well defined and the study was performed to high ethical standards. It is a strength of the study that all subjects had the assessments used for inputs into the predictive model.

However, there are a number of issues with the study design that significantly detract from the enthusiasm of the work in its current form. Most of these concerns stem from a lack of clarity about the clinical context of the patient encounters, the definition of the outcome that is being predicted, and the timing between the assessments and the outcomes. I will detail these concens below:

1) The study population was 'hospitalized PD patients'. This introduces tremendous variability into the model, since a patient hospitalized with urosepsis and multi-system organ failure in the ICU would be different than a PD patient who hospitalized with reffractory constipation, for example. How do the authors control for differences in the reason for admission?

The other issue with hospitalized settings is the differences in lengths of stay, which directly bears on the likelihood of the outcome (presumably, in hospital aspiration events). This is already demonstrated in Table 1 with longer stays in those with aspiration. How do the authors adjust the model for variability in the observation times? The model treats the outcome as a binary 'yes/no' occurrence, when it really is a event/time concept.

2) Another issue is that it is not clear when these assessments were performed, relative to the primary outcome. The model focuses on 3 major predictive factors, one of which is a single assessment (water swallow test) that seems quite sensitive to 'when' it was performed. Were all assessments performed on the same day of admission? Did any patients have repeated measures? The issue is that the results could be dynamic and an initially abnormal water swallow test could become normal (and vice versa) later in the admission. It also isn't clear how much time, on average (or median) elapsed between the abnormal water swallow test and the primary outcome of aspiration pneumonia.

3) It would seem likely that in the course of clinical care, that PD patients with an abnormal water swallow would have adjustments made in their diet (i.e., thickened liquids, etc.) in order to reduce the risk of aspiration. The predictive model does not seem to factor in dynamic clinical changes in diet / route of feeding that occurred prior to the outcome.

4) Finally, the definition of the primary study outcome -- 'aspiration' -- was not fully clear. Did it have to be an aspiration pneumonia? How did the authors know it was aspiration and not general hospital-acquired pneumonia? Some clues like right lower lobe consolidation are classic radiographic features that suggest aspiration, but the authors do not operationalize the definition of the primary study outcome. Please clarify this, as it is critical to be clear what is being predicted in this model.

Validity of the findings

As it currently stands, the authors do a good job of analyzing the data and using an appropriate method of analysis. The issue is that the clinical context for this model is on shaky footing -- 'hospitalized PD patients' is such a broad and varied context, that other factors related to the reason for hospitalization may be important -- these do not seem to have been factored into the model.

Additional comments

I would encourage the authors to define the variables more clearly, rather than dataset / annotated labels in Table 1. For example, it is a little unclear what CardCerebDis (cardiovascular cerebral disease) means as a Yes/No factor. The authors already have an exclusion criteria for patients with stroke. Does this mean carotid artery stenosis? If so, what percent is the cutoff for Yes/No? A similar logic can be used for many of the other variables -- weight loss (how much as cutoff for Yes/No?), pain (any pain anywhere?), etc.

Reviewer 2 ·

Basic reporting

1. Reads well and follows IMRaD format; background sufficiently explains clinical importance of aspiration in PD.

2. Table 1 mixes units/abbreviations (e.g., “HospitDur”). Use consistent units and spell out all abbreviations in a footnote.

3. State explicitly how the numeric cut-off of 89.56 points was mapped onto the graphical nomogram so that clinicians can reproduce the threshold.

Experimental design

1. The 25 % random split drawn from the same centre/time window does not test transportability. Provide validation on an external cohort (different hospital or later period) to demonstrate generalizability.

2. Sequential univariate screening → LASSO → forward stepwise logistic regression can inflate type-I error and shrinkage. Either justify this layered approach or (a) retain the LASSO-selected set with penalized coefficients, or (b) pre-specify clinically plausible variables and fit a ridge / shrinkage-corrected model.

3. Although events-per-variable is acceptable after selection (53 events / 3 predictors), n = 160 is small for nomogram derivation. Apply bootstrap internal validation (or k-fold cross-validation) and report optimism-adjusted calibration slope and intercept.

4. Specify whether any predictors or outcomes were missing and describe the strategy used (complete-case vs. multiple imputation).

5. Consent was verbal via telephone. Explain why written consent was not feasible and confirm that the IRB granted an explicit waiver of documentation.

6. Key aspiration-related factors (oral hygiene, saliva control, anticholinergic burden, cough-reflex sensitivity) were not collected. Discuss how these could be incorporated in future multi-centre studies.

Validity of the findings

1. Specificity is modest (0.68 training; 0.75 test), implying ≈ 25 % false positives at the Youden threshold. Discuss clinical consequences (e.g., unnecessary FEES) and consider whether a higher threshold would be preferable.

2. In the 40-patient test set, the NLR is reported as 0.00 because no false negatives occurred. Provide a 95 % confidence interval or label the metric as “not estimable” due to small sample size, and temper any claims of perfect rule-out.

3. Multi-step variable selection combined with a small single-centre sample heightens over-fitting. Without bootstrap adjustment or external validation, apparent discrimination and calibration may be optimistic.

4. Re-frame conclusions to acknowledge the model’s current limitation to one center and the need for external validation before clinical implementation.

---

## Round 0.2 · Minor Revisions

· Academic Editor

Minor Revisions

**Language Note:** When preparing your next revision, please ensure that your manuscript is reviewed either by a colleague who is proficient in English and familiar with the subject matter, or by a professional editing service. PeerJ offers language editing services; if you are interested, you may contact us at [email protected] for pricing details. Kindly include your manuscript number and title in your inquiry. – PeerJ Staff

·

Basic reporting

The revised manuscript is much improved in its description of the methodology.

Experimental design

The prediction is of early aspiration events (i.e., days "0-3" of the hospitalization), which does help refine the scope of the modeling exercise. The original manuscript was very unclear 'when' the aspiration events occurred. Although there may still be a conceptual issue in including patients "admitted with aspiration pneumonia" in the model that predicts that as an outcome, I believe enough patients developed aspiration events after admission to justify the model's translational value.

Validity of the findings

As above, there is still a potential problem, including patients admitted with aspiration in the model that predicts aspiration as an outcome. However, the model's performance characteristics and the fact that not all patients had aspiration on admission (i.e., the authors imply that many truly developed aspiration after assessment) suggest that this model would still function well, even if those admitted with aspiration were excluded from analysis.

Additional comments

I think it would help to add two pieces of information:

1) How many of the aspiration group were ADMITTED WITH ASPIRATION, as opposed to those who developed aspiration AFTER admission?

2) Add as a limitation --- the model's performance could be inflated by the inclusion of patients with the predicted outcome present at the time of the assessment/admission.

---

## Round 0.3 · accepted · Accept

· Academic Editor

Accept

The authors' have done well to address all the reviewers' comments. I have assessed the comments requested by Reviewer 2 and am happy with the revisions. This is ready for publication.

·

Basic reporting

No comment

Experimental design

The authors have clarified the design, which did NOT include those admitted with aspiration events. This seemed to me to be a pivotal concept not clearly stated in the initial iterations of the manuscript. It is explicitly clear in this version of the manuscript.

Validity of the findings

No Comment

Additional comments

The authors have done well to address concerns. I have no more comments about the current version of the manuscript.